# REPRESENTATION BOTTLENECK OF GRAPH NEURAL NETWORKS FOR SCIENTIFIC PROBLEMS

## ABSTRACT

Graph neural networks (GNNs) mainly rely on the message-passing paradigm to propagate node features and build interactions, and different graph learning problems require different ranges of node interactions. In this work, we explore the capacity of GNNs to capture node interactions under contexts of different complexities. We discover that *GNNs usually fail to capture the most informative kinds of interaction styles for diverse graph learning tasks*, and thus name this phenomenon as GNNs' representation bottleneck. As a response, we demonstrate that the inductive bias introduced by existing graph construction mechanisms can result in this representation bottleneck, *i.e.*, preventing GNNs from learning interactions of the most appropriate complexity. To address that limitation, we propose a novel graph rewiring approach based on interaction patterns learned by GNNs to adjust the receptive fields of each node dynamically. Extensive experiments on both real-world and synthetic datasets prove the effectiveness of our algorithm in alleviating the representation bottleneck and its superiority in enhancing the performance of GNNs over state-of-the-art graph rewiring baselines.

## 1 INTRODUCTION

Graph neural networks (GNNs) (Kipf & Welling, 2016; Hamilton et al., 2017) have witnessed growing popularity thanks to their ability to handle graphs that have complex relationships and interdependence between objects, ranging from social networks (Fan et al., 2019) to computer programs (Nair et al., 2020). Particularly, GNNs show promising strength in scientific research. They are used to derive insights from structures of molecules (Wu et al., 2018) and reason about relations in a group of interacting objects. Subsequently, significant efforts have been devoted to fully leveraging 3D geometry such as directions (Klicpera et al., 2020b;a) and dihedral angles (Klicpera et al., 2021; Liu et al., 2021). Since physical rules stay stable regardless of the reference coordinate system, equivariance has been regarded as a ubiquitous property and integrated upon GNNs with remarkable benefits (Ingraham et al., 2019; Fuchs et al., 2020; Hutchinson et al., 2021; Satorras et al., 2021).

The success of GNNs provokes the succeeding bottleneck question: "*What are the common limitations of GNNs in real-world modeling applications, such as molecules and dynamic systems?*" Since the majority of GNNs are expressed as a neighborhood aggregation or message-passing scheme (Gilmer et al., 2017; Veličković et al., 2017), we leverage the tool of multi-order interactions between input variables (Deng et al., 2021) to investigate their representation bottleneck, aiming to analyze which types of interaction patterns (*e.g.*, certain physical or chemical concepts) are likely to be encoded by GNNs, and which others are difficult to manipulate.

To this end, we formulate the metric of the multi-order interactions for GNNs from both node-level and graph-level perspectives. With this computational instrument, we observe that the distribution of different kinds of node interactions learned by GNNs can deviate significantly from the initial data distribution of interactions. Furthermore, we relate the capacity of GNNs to capture interactions under different complexities with their model expressiveness. It is discovered that *GNNs are typically incapable of learning the most informative interaction patterns and therefore cannot reach the global minimal loss point*, which we call the **representation bottleneck** of GNNs.

In this paper, we first attempt to explain this GNNs' representation bottleneck via graph construction mechanisms. We prove that existing methodologies to build node connectivity in scientific domains

such as *K-nearest neighbor* (KNN) and *fully connection* (FC) can introduce improper inductive bias. This improper inductive bias hidden inside the assumption of graph connectivity prohibits GNNs from encoding some particular interaction modes. To resolve the abovementioned obstacle, we propose a novel graph rewiring technique based on the distribution of interaction strengths named ISGR. It first detects the interaction pattern that changes most violently, which then is used to progressively optimize the inductive bias of GNNs and calibrate the topological structures of input graphs. Massive experimental evidence on both synthetic and real-world datasets validate its considerable potential to ameliorate the representation bottleneck and achieve stronger interpretability and generalization for GNNs against all graph rewiring baselines.

Last but not least, we revisit the representation behaviors of other categories of DNNs and compare them with GNNs. As a relevant answer, we observe the liability of CNNs to capture too simple pairwise interactions rather than more complex ones, which conforms to preceding discovery (Deng et al., 2021). This inclination of CNNs can also be explained by the inductive bias as CNNs' fairly small kernel size usually assumes local connections between pixels or patches. However, as opposed to GNNs, CNNs are far more vulnerable to and seldom diverge from the original data distribution of node interactions. This analysis illustrates that both similarities and significant discrepancies exist between GNNs and other classes of DNNs in their representation activities. Noticeably, due to the limitation of writing space, we moved the detailed literature review to Appendix E.

## 2 RELATED WORKS

**GNNs' expressiveness and bottlenecks.** GNNs are found to capture only a tiny fragment of first-order logic (Barceló et al., 2020), which arises from the deficiency of a node's receptive field. Meanwhile, GNNs do not benefit from the increase of layers due to *over-smoothing* (Li et al., 2018a; Klicpera et al., 2018; Chen et al., 2020) and *over-squashing* (Alon & Yahav, 2020; Topping et al., 2021). To the best of our knowledge, none considers GNNs' capacity in encoding pairwise interactions, and we are the foremost to understand GNNs' expressiveness from interactions under different contextual complexities and link the expressive limitation with the inductive bias of graph connectivity. More elaborate related works are in Appendix E.

**Graph rewiring.** Rewiring is a process of altering the graph structure to control the information flow. Among existing approaches such as connectivity diffusion (Klicpera et al., 2019), bridge-node insertion (Battaglia et al., 2018), and multi-hop filters (Frasca et al., 2020), edge sampling shows great power in tackling *over-smoothing* and *over-squashing*. The sampling strategies can be random drop (Huang et al., 2020b) or based on edge relevance (Klicpera et al., 2019; Kazi et al., 2022). For instance, Alon & Yahav (2020) modify the last layer to a FC-graph to help GNNs grab long-range interactions. Taking a step further, Topping et al. (2021) prove that negatively curved edges are responsible for *over-squashing* and introduces a curvature-based rewiring method to alleviate that. Differently, our rewiring algorithm originates from a completely new motivation, *i.e.*, reshaping graph structure to assist GNNs to learn the most informative order of interactions.

## 3 MULTI-ORDER INTERACTIONS FOR GRAPHS

**Preliminary for GNNs** Suppose a graph $\mathcal{G} = (\mathcal{V}, \mathcal{E})$ has a set of $n$ variables (*a.k.a.* nodes). $\mathcal{G}$ can be a macroscopic physical system with $n$ celestial bodies, or a microscopic biochemical system with $n$ atoms, denoted as $\mathcal{V} = \{v_1, ..., v_n\}$. $f$ is a well-trained GNN model and $f(\mathcal{G})$ represents the model output. For node-level tasks, the GNN forecasts a value (*e.g.*, atomic energy) or a vector (*e.g.*, atomic force or velocity) for each node. For graph-level tasks, $f(\mathcal{G}) \in \mathbb{R}$ is a scalar (*e.g.*, drug toxicity or binding affinity). Most GNNs make predictions by interactions between input nodes instead of working individually on each vertex (Qi et al., 2018; Li et al., 2019; Lu et al., 2019; Huang et al., 2020a). Accordingly, we concentrate on pairwise interactions and use the multi-order interaction $I^{(m)}(i, j)$ (Bien et al., 2013; Tsang et al., 2017) to measure interactions of different complexities between two nodes $v_i, v_j \in \mathcal{V}$.

**Graph-level Multi-order Interactions** Specifically, the $m$-th order interaction $I^{(m)}(i, j)$ measures the average interaction utility between nodes $v_i$ and $v_j$ under all possible subgraphs $\mathcal{G}_S$, which

consists of $m$ nodes. Mathematically, the multi-order interaction is defined as:

$$I^{(m)}(i,j) = \mathbb{E}_{\mathcal{G}_S \subseteq \mathcal{G}, \{v_i, v_j\} \subseteq \mathcal{V}, |\mathcal{V}_S| = m}[\Delta f(v_i, v_j, \mathcal{G}_S)], \qquad (1)$$

where $3 \le m \le n$ and $\Delta f(v_i, v_j, \mathcal{G}_S)$ is defined as $f(\mathcal{G}_S) - f(\mathcal{G}_S \backslash v_i) - f(\mathcal{G}_S \backslash v_j) + f(\mathcal{G}_S \backslash \{v_i, v_j\})$. $\mathcal{G}_S \subset \mathcal{G}$ is the context subgraph. $f(\mathcal{G}_S)$ is the GNN output when we keep nodes in $\mathcal{G}_S$ unchanged but delete others in $\mathcal{G} \backslash \mathcal{G}_S$. Since it is irrational to feed an empty graph into a GNN, we demand the context $S$ to have at least one variable with $m \ge 3$ and omit $f(\emptyset)$. Note that some studies (Zhang et al., 2020) assume the target variables $v_i$ and $v_j$ do not belong to the context $\mathcal{G}_S$. Contrarily, we propose to interpret $m$ as the contextual complexity of the interaction and include nodes $v_i$ and $v_j$ in the subgraph $\mathcal{G}_S$. Proof is provided in Appendix A.2 that the two cases are equivalent but from different views. An elaborate introduction of $I^{(m)}$ (*e.g.*, the connection with existing metrics) is in Appendix A.

**Node-level Multi-order Interaction**   $I^{(m)}(i,j)$ in Equ. 1 is designed to analyze the influence of interactions over the integral system (*e.g.*, a molecule or a galaxy) and is therefore only suitable in the circumstance of graph-level prediction. However, many graph learning problems are node-level tasks (Zachary, 1977). For the sake of measuring the effects of those interactions on each component (*e.g.*, atom or particle) of the system, we introduce a new metric as the following:

$$I_i^{(m)}(j) = \mathbb{E}_{\mathcal{G}_S \subseteq \mathcal{G}, v_j \in \mathcal{V}, |\mathcal{V}_S| = m}[\Delta f_i(v_j, \mathcal{G}_S)], \qquad (2)$$

where $2 \le m \le N$ and $\Delta f_i(v_j, \mathcal{G}_S)$ is formulated as $\|f_i(\mathcal{G}_S) - f_i(\mathcal{G}_S \backslash v_j)\|_p$. $\|.\|_p$ is the $p$-norm. We denote $f_i(\mathcal{G}_S)$ as the output for node $v_i$ when other nodes in $\mathcal{G}_S$ are kept unchanged. Equ. 2 allows us to measure the representation capability of GNNs in node-level tasks.

**Interaction Strengths**   To measure the reasoning complexity of GNNs, we compute the graph-level relative interaction strength $J^{(m)}$ of the encoded $m$-th order interaction as follows:

$$J^{(m)} = \frac{\mathbb{E}_{\mathcal{G} \in \Omega}\left[\mathbb{E}_{i,j}\left[|\ I^{(m)}(i,j \mid \mathcal{G})\ |\right]\right]}{\sum_{m'}\left[\mathbb{E}_{\mathcal{G} \in \Omega}\left[\mathbb{E}_{i,j}\left[|\ I^{(m')}(i,j \mid \mathcal{G})\ |\right]\right]\right]}, \qquad (3)$$

where $\Omega$ stands for the set of all graph samples, and the strength $J^{(m)}$ is calculated over all pairs of input variables in all data points. Remarkably, the distribution of $J^{(m)}$ measures the distribution of the complexity of interactions encoded in GNNs. Then we normalize $J^{(m)}$ by the summation value of $I^{(m)}(i,j \mid \mathcal{G})$ with different orders to constrain $0 \le J^{(m)} \le 1$ for explicit comparison across various tasks and datasets. Then the node-level interaction strength is defined as $J^{(m)} = \frac{\mathbb{E}_{\mathcal{G} \in \Omega}\left[\mathbb{E}_i\left[\mathbb{E}_j\left[|I_i^{(m)}(j|\mathcal{G})|\right]\right]\right]}{\sum_{m'}\left[\mathbb{E}_{\mathcal{G} \in \Omega}\left[\mathbb{E}_i\left[\mathbb{E}_j\left[|I_i^{(m')}(j|\mathcal{G})|\right]\right]\right]\right]}$.

## 4 REPRESENTATION BOTTLENECK OF GNNS

### 4.1 DEFINITION OF REPRESENTATION BOTTLENECK

For modern GNNs, the loss $L$ is typically non-convex with multiple local and even global minima (Foret et al., 2020). This diversity of minima may yield similar values of $L$ while acquiring different capacities of GNNs to learn interactions. That is, each loss point must correspond to different $J^{(m)}$. Intuitively, we declare in Prop. 1 that, if $J^{(m)}$ learned by a GNN model $f$ is not equivalent to the optimal strength $J^{(m)^*}$, then $f$ must be stuck in a local minimum point of the loss surface (explanation is in Appendix A.4).

**Proposition 1** *Let $J^{(m)^*}$ be the interaction strength of GNN $f^*$ that achieves the global minimum loss $\ell^*$ on some data $D$. If another GNN model $f'$ converges to a loss $\ell'$ after the parameters update and $J^{(m)'} \ne J^{(m)^*}$, then $\ell'$ must be a local minimum loss, i.e, $\ell' > \ell^*$.*

Notably, different tasks require diverse ranges of interactions. For instance, short-range interactions play a more critical role in small-molecule binding prediction (Wu et al., 2022a), while protein-protein rigid-body docking attaches more significance to the middle-range or long-range interactions

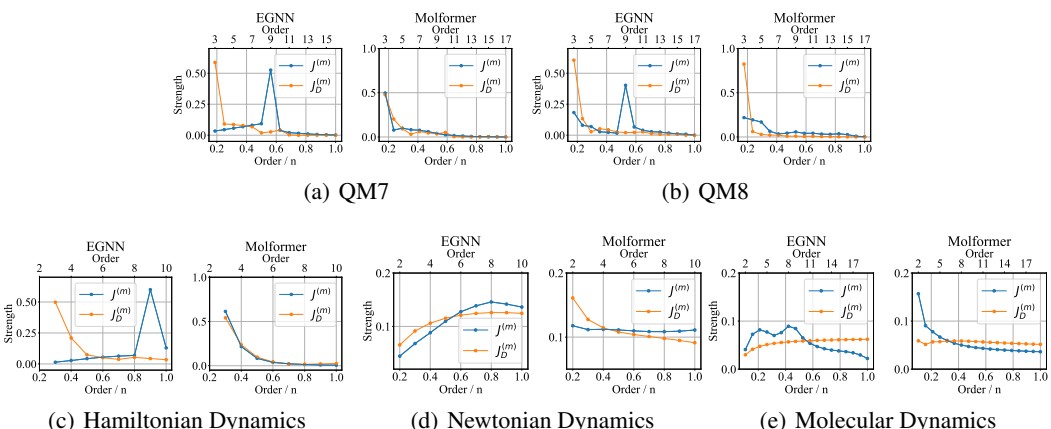

Figure 1: Distributions of interaction strengths of EGNN and Molformer in graph-level and node-level prediction tasks. We use double-x axes to represent the order $m$ and the ratio $m/n$

between residues (Ganea et al., 2021). Thus, $J^{(m)^*}$ varies according to particular tasks and datasets. Based on the relationship between $J^{(m)}$ and the training loss $l$ described in Prop. 1, we define the representation bottleneck of GNNs as the phenomenon that *the distribution of interaction strength $J^{(m)}$ learned by GNNs fails to reach the optimal distribution of interaction strength $J^{(m)^*}$.*

## 4.2 THE ROLE OF INDUCTIVE BIAS

Many factors can instigate this GNNs' representation bottleneck. In this paper, we focus on the improper inductive bias introduced by the graph construction mechanism as a partial answer.

**GNNs for Scientific Problems**   GNNs operate on graph-structured data and have strong ties to the field of geometric deep learning. Aside from studies on social networks Fan et al. (2019); Hamilton et al. (2017) and citation networks Sen et al. (2008); Tang et al. (2008) as well as knowledge graphs Carlson et al. (2010), science including biology, physics, and chemistry has been one of the main drivers in the development of GNNs (Battaglia et al., 2016; Li et al., 2018b; Mrowca et al., 2018; Sanchez-Gonzalez et al., 2020; Wu et al., 2022b). In this work, our concentration is put on analyzing the representation behavior of geometric GNNs for scientific explorations.

Notably, graphs in most scientific problems are unlike common applications of GNNs such as recommendation systems (Wu et al., 2022c) and relation extraction (Zhu et al., 2019). Indeed, molecules or dynamic systems do not have explicit edges. *KNN*, *full connections* (FC), and *r-ball* are the three most broadly used mechanisms to build node connectivity. KNN-graphs build edges based on pairwise distances in the 3D space and are a common technique in modeling macromolecules (Fout et al., 2017; Ganea et al., 2021; Stärk et al., 2022). FC-graphs, instead, assume all nodes are connected to each other (Chen et al., 2019; Wu et al., 2021; Baek et al., 2021; Jumper et al., 2021). In *r*-ball graphs, an edge between any node pair exists as long as their spatial distance is shorter than a threshold value.

In order to align with the multi-order interaction theory, graph construction must satisfy two properties: (1) The subgraph maintains the connectivity. (2) No ambiguity is intrigued from either the structural or feature view. However, *r*-ball graphs do not satisfy the first property. If node $a$ is merely linked to node $b$, it would be isolated if $b$ is removed from the graph. In contrast, subgraphs in KNN-graphs can be re-constructed via KNN to ensure connectivity, while the removal of any node in FC-graphs will not influence the association of other entity pairs. Hence, we only consider KNN- and FC-graphs in the subsequent analysis.

**Inductive Bias of GNNs**   We define the data distribution of interaction strengths on a dataset $D$, denoted as $J_D^{(m)}$, as the experimental distribution of interaction strengths for model $f$ with randomly initialized parameters, which is later used for the comparison between GNNs and other DNNs. Fig. 1

Table 1: Comparison of graph rewiring methods for graph-level and node-level prediction tasks.

| Task Model | Hamiltonian Dynamics | | QM7 | | QM8 | | Newtonian Dynamics | | Molecular Dynamics | |
|---|---|---|---|---|---|---|---|---|---|---|
| | EGNN | Molformer | EGNN | Molformer | EGNN | Molformer | EGNN | Molformer | EGNN | Molformer |
| None | $1.392 \pm 0.042$ | $1.545 \pm 0.036$ | $68.182 \pm 3.581$ | $51.119 \pm 2.193$ | $0.012 \pm 0.001$ | $0.012 \pm 0.001$ | $6.951 \pm 0.098$ | $1.929 \pm 0.051$ | $1.409 \pm 0.082$ | $0.848 \pm 0.053$ |
| +FA | $1.168 \pm 0.043$ | – | $55.288 \pm 3.074$ | – | $0.012 \pm 0.001$ | – | $5.348 \pm 0.183$ | – | $0.826 \pm 0.105$ | – |
| DIGL | $1.151 \pm 0.044$ | $1.337 \pm 0.072$ | $61.028 \pm 3.804$ | $41.188 \pm 5.329$ | $0.012 \pm 0.001$ | $0.011 \pm 0.001$ | $5.637 \pm 0.147$ | $1.902 \pm 0.081$ | $1.108 \pm 0.131$ | $0.790 \pm 0.078$ |
| SDRF | $1.033 \pm 0.790$ | $1.265 \pm 0.039$ | $59.921 \pm 3.765$ | $35.792 \pm 4.565$ | $0.011 \pm 0.001$ | $0.011 \pm 0.001$ | $5.460 \pm 0.133$ | $1.885 \pm 0.068$ | $0.942 \pm 0.152$ | $0.751 \pm 0.046$ |
| ISGR | $\mathbf{0.892 \pm 0.051}$ | $\mathbf{1.250 \pm 0.029}$ | $\mathbf{53.134 \pm 2.711}$ | $\mathbf{34.439 \pm 4.017}$ | $\mathbf{0.011 \pm 0.000}$ | $\mathbf{0.010 \pm 0.001}$ | $\mathbf{4.734 \pm 0.103}$ | $\mathbf{1.879 \pm 0.066}$ | $\mathbf{0.713 \pm 0.097}$ | $\mathbf{0.736 \pm 0.048}$ |

reports the learned distributions $J^{(m)}$ and the data distributions $J_D^{(m)}$ for both graph-level and node-level tasks, where EGNN (Satorras et al., 2021) works on KNN-graphs and Molformer (Wu et al., 2021) performs on FC-graphs. The complete experimental details are elaborated in subsection 4.3.

Based on these empirical plots, it can be observed that EGNN and Molformer are inclined to learn interactions of distinct orders. To be specific, EGNN is more prone to pay attention to interactions of some particular orders, and its $J^{(m)}$ lines usually have one or two spikes. Contrarily, Molformer learns a more unconstrained $J^{(m)}$. Its $J^{(m)}$ on Newtonian dynamics is extremely smooth like a straight line, but its $J^{(m)}$ on Hamiltonian and molecular dynamics (MD) are steep curves.

We owe this phenomenon to different inductive biases brought by different graph construction approaches. Namely, the inductive bias brought by the topological structure of input graphs significantly impacts $J^{(m)}$ of GNNs. Noticeably, we demonstrate in Sec. 6 that CNNs can be forbidden from capturing appropriate orders of interactions due to their inductive bias of locality. This local characteristic comes from CNNs' relatively small kernel size. As for GNNs, their representation manners can also be influenced by the inductive bias, which primarily depends on graph connectivity. Unequivocally, FC-graphs consist of all pairwise relations and hypothesize that all particles can affect each other directly. This hypothesis places weaker restrictions on Molformer's representation behavior and results in more diverse $J^{(m)}$. On the contrary, KNN-graphs assume that some pairs of entities possess a connection and others do not. In our setting, the number of nearest neighbors $K$ for KNN-graphs is kept as $K = 8$, which is equivalent to the order that has the highest strength in $J^{(m)}$ of EGNN.

Undeniably, inductive biases introduced by graph construction mechanisms can be improper, and, subsequently, give rise to poor $J^{(m)}$ that is far away from $J^{(m)*}$ after training. Prominently, bad inductive bias can impose a much huger impact for GNNs on $J^{(m)}$ than for CNNs. This is because graphs support arbitrary pairwise relational structures (Battaglia et al., 2018) and accordingly, the inductive bias of GNNs is more flexible and influential. Especially, KNN-graphs are more susceptible to improper inductive bias. EGNN hardly learns interactions of orders different from the user-defined constant $K$, which can cause worse performance. However, FC-graphs are not panacea. On the one hand, FC-graphs require far more computational costs and may be prohibited in the case of tremendous entities. On the other hand, the performance of Molformer severely depends on the sufficiency and quality of training data. As shown in Tab. 1, Molformer does not surpass EGNN on all datasets and instead behaves worse than EGNN on Hamiltonian ($1.250 > 0.892$) and MD ($0.736 > 0.713$).

## 4.3 IMPLEMENTATION DETAILS

### 4.3.1 TASKS AND DATASETS

We present four tasks to explore the representation patterns of GNNs for scientific research. Among them, molecular property prediction and Hamiltonian dynamics are graph-level prediction tasks, while Newtonian dynamics and molecular dynamics simulations are node-level ones.

**Molecular property prediction.** The forecast of a broad range of molecular properties is a fundamental task in drug discovery (Drews, 2000). The properties in current molecular collections can be mainly divided into four categories: quantum mechanics, physical chemistry, biophysics, and physiology, ranging from molecular-level properties to macroscopic influences on the human body (Wu et al., 2018). We utilize two benchmark datasets. QM7 (Blum & Reymond, 2009) is a subset of GDB-13 and is composed of 7K molecules. QM8 (Ramakrishnan et al., 2015) is a subset of GDB-

17 with 22K molecules. Note that QM7 and QM8 provide one and twelve properties, respectively, and we merely use the *E1-CC2* property in QM8 for simplicity.

**Newtonian dynamics.** Newtonian dynamics (Whiteside, 1966) describes the dynamics of particles according to Newton's law of motion: the motion of each particle is modeled using incident forces from nearby particles, which changes its position, velocity, and acceleration. Several important forces in physics, such as the gravitational force, are defined on pairs of particles, analogous to the message function of GNNs (Cranmer et al., 2020). We adopt the N-body particle simulation dataset in (Cranmer et al., 2020). It consists of N-body particles under six different interaction laws. More details can be referred to Appendix B.1.

**Hamiltonian dynamics.** Hamiltonian dynamics (Greydanus et al., 2019) describes a system's total energy $\mathcal{H}(\mathbf{q}, \mathbf{p})$ as a function of its canonical coordinates $\mathbf{q}$ and momenta $\mathbf{p}$, *e.g.*, each particles' position and momentum. The dynamics of the system change perpendicularly to the gradient of $\mathcal{H}$: $\frac{d\mathbf{q}}{dt} = \frac{\partial \mathcal{H}}{\partial \mathbf{p}}, \frac{d\mathbf{p}}{dt} = -\frac{d\mathcal{H}}{d\mathbf{q}}$. There we take advantage of the same datasets from the Newtonian dynamics case study, and attempt to learn the scalar total energy $\mathcal{H}$ of the system.

**Molecular dynamics simulations.** MD (Frenkel & Smit, 2001; Karplus & McCammon, 2002) has long been the *de facto* choice for modeling complex atomistic systems from first principles. We adopt ISO17 (Schütt et al., 2017; 2018), which has 129 molecules and is generated from MD simulations using the Fritz-Haber Institute *ab initio* simulation package (Blum et al., 2009). Each molecule contains 5K conformational geometries and total energies with a resolution of 1 femtosecond in the trajectories. We predict the atomic forces in molecules at different timeframes.

### 4.3.2 OTHER EXPERIMENTAL SETTINGS

Two state-of-the-art geometric GNNs are selected to perform on these two graph types. We pick up Equivariant Graph Neural Network (EGNN) (Satorras et al., 2021) for KNN-graphs, and Molformer (Wu et al., 2021) with no motifs for FC-graphs. EGNN is roto-translation and reflection equivariant without the spherical harmonics (Thomas et al., 2018). Molformer is a variant of Transformer (Vaswani et al., 2017; Hernández & Amigó, 2021), designed for molecular graph learning. More experimental details are elucidated in Appendix B.

Meanwhile, it also needs to be clarified how to compute node features for $\mathcal{G} \backslash \mathcal{G}_S$. Noteworthily, the widely-used setting proposed by Ancona et al. (2019) for sequences or pixels is not applicable to GNNs, because an average over graph instances can lead to an ambiguous node type. Alternatively, we consider dropping nodes and corresponding edges in $\mathcal{G} \backslash \mathcal{G}_S$ instead of replacing them with a mean value.

## 5 INTERACTION STRENGTH-BASED GRAPH REWIRING

### 5.1 REWIRING FOR INDUCTIVE BIAS OPTIMIZATION

In order to approach $J^{(m)^*}$, recent work (Deng et al., 2021) imposes two losses to encourage or penalize learning interactions of specific complexities. Nevertheless, they require models to make accurate predictions on subgraphs. But variable removal brings the out-of-distribution (OOD) problem (Chang et al., 2018; Frye et al., 2020; Wang et al., 2022), which can manipulate GNNs' outcome arbitrarily and produce erroneous predictions (Dai et al., 2018; Zügner et al., 2018). More importantly, these losses are based on the assumption that the image class remains regardless of pixel removal. But it is not rational to assume the stability of graph properties if we alter their components. In this work, instead of intervening in the loss, we rely on the modification of GNNs' inductive bias to capture the most informative order $m^*$ of interactions and therefore reach $J^{(m)^*}$.

Unfortunately, $m^*$ can never be known unless sufficient domain knowledge is supplied. As shown in Fig. 3, $J^{(m)}$ for CNNs does not directly dive into low-order of interactions in the initial training epochs (*e.g.*, 10 or 50 epochs). Instead, CNNs have the inclination to learn a more informative order of interactions (*e.g.*, middle-order) regardless of the inductive bias, which has also been observed

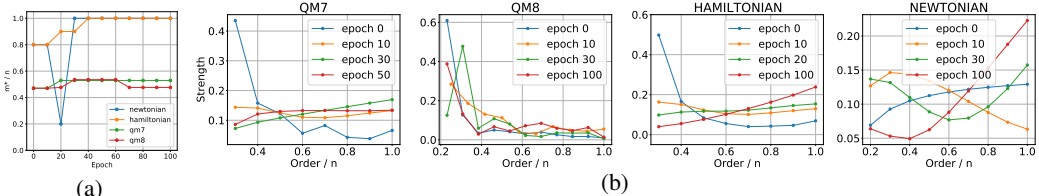

Figure 2: (a) The change of $m^*$ over epochs for EGNN. (b) The change of interaction strengths over epochs for EGNN.

for GNNs. So, motivated by this subtle tendency, we resort to the order of interactions that increase the most in $J^{(m)}$ during training as the guidance to reconstruct graphs and estimate $J^{(m)^*}$.

To this end, we dynamically adjust the reception fields of each node within graphs by establishing or destroying edges based on the interaction strength $J^{(m)}$. To begin with, we randomly sample a mini-batch and calculate its corresponding interaction strengths. If the maximum increase of some order, *i.e.*, $\max_m \left( \Delta J_t^{(m)} - J_{t-1}^{(m)} \right)$, exceeds the pre-defined threshold $\bar{J}$, then we modify the number of nearest neighbors $K$ to be closer to the order $m^*$ whose interaction strength rises the most as $K = (m^* + K)/2$. After that, we reconstruct the node connectivity for $\{\mathcal{G}_i\}_{i=1}^N$ according to the new $K$. This process is iteratively implemented every $\Delta e$ epoch for an efficient training speed.

Such a method is often generically referred to as *graph rewiring* (Topping et al., 2021), so it is dubbed Interaction Strength-based Graph Rewiring (ISGR) as described in Alg. 1. By adjusting the graph topology that arouses the inductive bias, GNNs are enabled to break the representation bottleneck to some extent and $J^{(m)}$ is able to gradually approximate $J^{(m)^*}$.

## 5.2 EFFECTS OF ISGR ALGORITHM

**Baselines.** We compare ISGR to a variety of graph rewiring methods. **+FA** (Alon & Yahav, 2020) modifies the last GNN layer to be fully connected. **DIGL** (Klicpera et al., 2019) leverages generalized graph diffusion to smooth out the graph adjacency and promote connections among nodes at short diffusion distances. **SDRF** (Topping et al., 2021) is the state-of-the-art rewiring technique and alleviates a graph's strongly negatively curved edges.

**Results.** Extensive experiments are conducted to examine the efficiency of our ISGR method. Tab. 1 documents the outcome with the mean and standard deviation of three repetitions, where the top two are in bold and underlined, respectively. Our ISGR algorithm significantly improves the performance of EGNN and Molformer upon all baselines on both graph-level and node-level tasks. Particularly, the promotion of ISGR for EGNN is much higher, which confirms our assertion that GNNs based on KNN-graphs are more likely to suffer from bad inductive bias. On the flip side, the improvement for Molformer in QM7 is more considerable than in QM8. This proves that GNNs based on FC-graphs are more easily

---

**Algorithm 1** The workflow of ISGR Algorithm.

**Input:** threshold $\bar{J}$, total epochs $E$, interval $\Delta e$

**for** $t = 1, 2, ..., E/\Delta e$ **do**
  $J_t^{(m)} \leftarrow$ calculate Equ. 1 on a random batch
  **if** $\max_m \left( J_t^{(m)} - J_{t-1}^{(m)} \right) > \bar{J}$ **then**
    $m^* \leftarrow \arg\max_m \left( J_t^{(m)} - J_{t-1}^{(m)} \right)$
    $K \leftarrow \frac{m^*+K}{2}$    ▷ make $K$ closer to $m^*$
    $\{\mathcal{G}_i\}_{i=1}^N \leftarrow K$   ▷ reset $K$ for all KNN-graphs
  **end if**
**end for**

---

affected by inappropriate inductive bias (*i.e.*, full connection) when the data is insufficient since the size of QM7 (7K) is far smaller than QM8 (21K). We also see that +FA outweighs DIGL and SDRF, the rewiring algorithms by edge sampling, when the graph connectivity is built on KNN. However, when encountering FC-graphs, +FA losses efficacy, and SDRF achieves a larger improvement than DIGL.

**Change of $m^*$.** Fig. 2 plots the variation tendency of $m^*$ over epochs, showing that different tasks enjoy various optimal $K$ (denoted as $K^*$). Explicitly, Hamiltonian dynamics and Newtonian

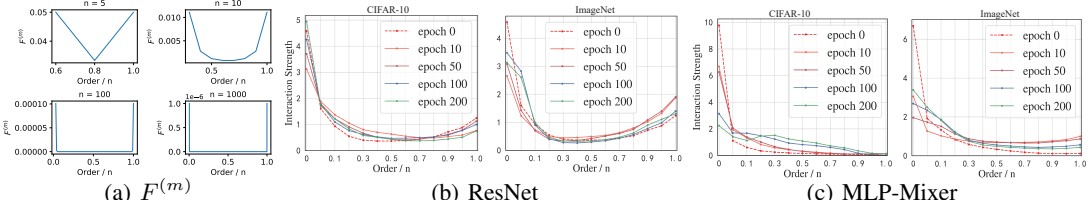

Figure 3: (a) The theoretical distributions of $F^{(m)}$ under different $n$. (b) The change of interaction strengths for ResNet on CIFAR-10 and ImageNet. (c) The change of interaction strengths for MLP-Mixer on CIFAR-10 and ImageNet.

dynamics benefit from the full connection ($K^*/n = 1$), while the molecular property prediction benefits more from middle-order interactions ($K^*/n \approx 0.5$). This phenomenon perfectly fits the physical laws, because the system in Newtonian and Hamiltonian datasets is extremely compact with close pairwise distances. Those particles are more likely to be influenced by all the other nodes.

**The change of interaction strengths during training.** Fig. 2 also depicts how $J^{(m)}$ changes when the training proceeds with our ISGR algorithm. Although for molecular property prediction and Hamiltonian dynamics, $J_D^{(m)}$ mostly concentrate on low-order interactions ($m/n \leq 0.4$), $J^{(m)}$ progressively adjust to middle- and high-order ($m/n \geq 0.4$). Regarding Newtonian dynamics, $J_D^{(m)}$ is very smooth, but $J^{(m)}$ at initial epochs (*i.e.*, 10 and 20 epochs) oddly focus on low-order interactions ($m/n \leq 0.4$). Nevertheless, our ISGR method timely corrects the wrong tendency, and eventually, $J^{(m)}$ becomes more intensive in segments of middle- and high-order ($m/n \geq 0.6$).

## 6 REPRESENTATION BOTTLENECKS OF DNNS

### 6.1 REVISITING FINDINGS OF OTHER DNNS

Recently, the representation bottleneck of other DNNs (*e.g.*, CNNs) has been burgeoningly investigated. Some (Deng et al., 2021) leverage $\Delta W^{(m)}(i, j) = R^{(m)} \frac{\partial I^{(m)}(i,j)}{\partial W}$ in Equ. **??** to represent the compositional component of $\Delta W$ w.r.t. $\partial I^{(m)}(i, j)/\partial W$. They claim that $| \Delta W^{(m)}(i, j) |$ is proportional to $F^{(m)} = \frac{n-m+1}{n(n-1)}/\sqrt{\binom{n-2}{m-2}}$ and experimentally show that $J^{(m)}$ of low-order (*e.g.*, $m = 0.05n$) is much higher than high-order (*e.g.*, $m = 0.95n$).

Despite their fruitful progress, we argue that $F^{(m)}$ ought to be approximately the same when $m/n \to 0$ or $m/n \to 1$ (see Fig. 3) if the empty set $\emptyset$ is excluded from the input for DNNs. In Appendix A.3, we demonstrate that even if $f(\emptyset)$ is taken into consideration and $n$ is large (*e.g.*, $n \geq 100$), $J^{(m)}$ ought to be non-zero only when $m/n \to 0$. Therefore, if $| \Delta W^{(m)}(i, j) |$ depends entirely on $F^{(m)}$, DNNs should fail to capture any middle- or high-order interactions. But DNNs have performed well in tasks that require high-order interactions such as protein-protein rigid-body docking and protein interface prediction (Liu et al., 2020).

The inaccurate statement that $F^{(m)}$ determines $| \Delta W^{(m)}(i, j) |$ is due to the flawed hypothesis that the derivatives of $\Delta f(i, j, S)$ over model parameters, *i.e.*, $\partial \Delta f(i, j, S)/\partial W$, conform to normal distributions (see Appendix B.3). $\partial \Delta f(i, j, S)/\partial W$, undoubtedly, varies with the contextual complexities (*i.e.*, $|S|$), and replies on not only the data distribution of interaction strengths in particular datasets $J_D^{(m)}$ but the model architecture $f$.

### 6.2 INDUCTIVE BIAS IN OTHER DNNS

To verify our dissent, we reproduce the interaction strengths of ResNet in CIFAR-10 (Krizhevsky et al., 2009) and ImageNet (Russakovsky et al., 2015) in Fig. 3. The plot implies that $J_D^{(m)}$'s (referring to the epoch-0 curve) interactions of low and high orders are much stronger than middle orders. As previously analyzed, this phenomenon is due to a critical inductive bias of CNNs, *i.e.*,

locality. It assumes that entities are in spatially close proximity with one another and isolated from distant ones (Battaglia et al., 2018). Hence, CNNs with small kernel sizes are bound to low-order interactions.

Several recent studies have demonstrated that increasing the kernel size can alleviate the local inductive bias (Ding et al., 2022). Based on their revelation, we examine the change of interaction strengths for MLP using MLP-Mixer (Tolstikhin et al., 2021) in Fig. 3. Though MLP shares a similar $J_D^{(m)}$ with ResNet, its $J^{(m)}$ is much smoother. This is because MLP-Mixer assumes full connection of different patches with no constraint of the locality. Therefore, it can learn a more adorable $J^{(m)}$. The implementation details on visual tasks are elaborated in Appendix C.

### 6.3 Comparison between CNNs and GNNs

Recall that in Fig. 1, $J^{(m)}$ learned by GNNs can deviate from and is nearly independent of the data distribution $J_D^{(m)}$. Precisely, $J_D^{(m)}$ for molecular property prediction in QM8 is more intensive on low orders. But after sufficient training, $J^{(m)}$ for EGNN mainly have high values for middle orders. $J^{(m)}$ for Molformer also increases the most in the middle-order segment. The trend of $J^{(m)}$ illustrates that subgraphs with a middle size are exceedingly informative substructures that reveal small molecules' biological or chemical properties. This finding verifies that motifs such as functional groups heavily determine molecular attributes (Yu et al., 2020; Wang et al., 2021; Wu et al., 2022d). Similarly, although $J_D^{(m)}$ in QM7 concentrate on low-order, its $J^{(m)}$ are mainly allocated on middle-order. Especially for EGNN, its spike of $J^{(m)}$ is at $m = 9$. Concerning Molformer, the segment of its $J^{(m)}$ that increases most is dispersive between $0.3 \leq m/n \leq 0.5$. While for Hamiltonian dynamic systems, $J_D^{(m)}$ is majorly intense for low and middle orders. In spite of that, $J^{(m)}$ of EGNN concentrates more on high orders but neglects low orders. Regarding node-level prediction tasks, the scenery is more straightforward. $J_D^{(m)}$ for EGNN and Molformer are in different shapes, but $J^{(m)}$ both move towards low orders for MD and high-order interactions for Newtonian dynamics. All those phenomenons bolster a considerable discrepancy between $J^{(m)}$ and $J_D^{(m)}$ for geometric GNNs.

In opposition, as shown in Fig. 3, little difference exists between $J_D^{(m)}$ and $J^{(m)}$ (referring to non-zero epoch curves) at different epochs for both ResNet and MLP-Mixer. That is, $J^{(m)}$ learned by other sorts of DNNs, such as CNNs, are seldom divergent from $J_D^{(m)}$. This fact supports our previous claim in Sec. 4 that graph learning problems are more affected by inductive bias than visual tasks.

## 7 Conclusions and Limitations

We discover and strictly analyze the representation bottleneck of GNNs from the complexity of interactions encoded in networks. Remarkably, inductive bias rather than the data distribution is more dominant in the expressions of GNNs to capture pairwise interactions, contradicting the behavior of other DNNs. Besides, empirical results demonstrate that inductive biases introduced by most graph construction mechanisms, such as KNN and full connection, can be sub-optimal and prevent GNNs from learning the most revelatory order of interactions. Inspired by this gap, we design a novel rewiring method based on the inclination of GNNs to encode more informative interactions.

**Limitations and Future Directions**   Despite the reported success of our proposed ISGR, GNNs still face a few limitations. In real-world applications such as social networks (Hamilton et al., 2017; Fan et al., 2019), improper modeling of interactions may lead to the failure to classify fake users and posts. In addition, since GNNs have been extensively deployed in assisting biologists with the discovery of new drugs, inappropriate modeling can postpone the screening progress and impose negative consequences on the drug design. Another limitation of our work is that results are only demonstrated on small systems with few particles and two sorts of GNNs. The generalization of our claim remains investigated on larger datasets and more advanced architectures.

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

## A  INTRODUCTION AND THEORETICAL ANALYSIS OF MULTI-ORDER INTERACTIONS

### A.1  INTRODUCTION OF MULTI-ORDER INTERACTIONS

In this subsection, we give a more detailed introduction to the multi-order interaction, which is employed to analyze the representation ability of GNNs and CNNs in the main body, in case some audiences may find it hard to follow. This introduction largely utilizes previous studies (Zhang et al., 2020; Deng et al., 2021) for reference, and we strongly recommend interesting readers take a glance at these articles.

Here we follow the same mathematical description of multi-order interactions as Deng et al. (2021) for better alignment and convenient comparison. $f$ is a pre-trained DNN (*e.g.*, CNN, RNN, Transformer, GNN), and an input sample (*e.g.*, an image or a text) has $n$ variables (*e.g.*, pixels or words) denoted as $N = \{1, ..., n\}$. Then the original multi-order interaction between input variables is originally defined as follows:

$$I^{(m)}(i,j) = \mathbb{E}_{S \subseteq N \setminus \{i,j\}, |S|=m}[\Delta f(i,j,S)], 3 \le m \le n, \tag{4}$$

where $\Delta f(i,j,S) = f(S \cup \{i,j\}) - f(S \cup \{i\}) - f(S \cup \{j\}) + f(S)$ and $S \subset N$ represents the context with $m$ variables. $I^{(m)}(i,j)$ denotes the interaction between variables $i,j \in N$ of the $m$-th order, which measures the average interaction utility between $i,j$ under contexts of $m$ variables. There are five desirable properties that $I^{(m)}(i,j)$ satisfies:

- **Linear property.** If two independent games $f_1$ and $f_2$ are combined, obtaining $g(S) = f_1(S) + f_2(S)$, then the multi-order interaction of the combined game is equivalent to the sum of multi-order interactions derived from $f_1$ and $f_2$, *i.e.*,$I_g^{(m)}(i,j) = I_{f_1}^{(m)}(i,j) + I_{f_2}^{(m)}(i,j)$.

- **Nullity property.** If a dummy variable $i \in N$ satisfies $\forall S \subseteq N \setminus \{i\}, f(S \cup \{i\}) = f(S) + f(\{i\})$, then variable $i$ has no interactions with other variables, *i.e.*, $\forall m, \forall j \in N \setminus \{i\}, I^{(m)}(i,j) = 0$.

- **Commutativity property.** Intuitively, $\forall i, j \in N, I^{(m)}(i,j) = I^{(m)}(j,i)$.

- **Symmetry property.** Suppose two variables $i, j$ are equal in the sense that $i, j$ have same co-operations with other variables, *i.e.*, $\forall S \subseteq N \setminus \{i,j\}, f(S \cup \{i\}) = f(S \cup \{j\})$, then we have $\forall k \in N, I^{(m)}(i,k) = I^{(m)}(j,k)$.

- **Efficiency property (Deng et al., 2021).** The output of a DNN can be decomposed into the sum of interactions of different orders between different pairs of variables as:

$$f(N) - f(\emptyset) = \sum_{i \in N} \mu_i + \sum_{i,j \in N, i \ne j} \sum_{m=0}^{n-2} w^{(m)} I^{(m)}(i,j), \tag{5}$$

where $\mu_i = f(\{i\}) - f(\emptyset)$ represents the independent effect of variable $i$, and $w^{(m)} = \frac{n-1-m}{n(n-1)}$.

**Connection with Shapley value and Shapley interaction index.** Shapley value is introduced to measure the numerical importance of each player to the total reward in a cooperative game, which has been widely accepted to interpret the decision of DNNs in recent years (Lundberg & Lee, 2017; Ancona et al., 2019). For a given DNN and an input sample with a set of input variables $N = \{1, \ldots, n\}$, we use $2^N = \{S \mid S \subseteq N\}$ to denote all possible variable subsets of $N$. Then, DNN $f$ can be considered as $f : 2^N \to \mathbb{R}$ that calculates the output $f(S)$ of each specific subset $S \subseteq N$. Each input variable $i$ is regarded as a player, and the network output $f(N)$ of all input variables can be considered as the total reward of the game. The Shapley value aims to fairly distribute the network output to each individual variable as follows:

$$\phi_i = \sum_{S \subseteq N \setminus \{i\}} \frac{\mid S \mid !(n- \mid S \mid -1)!}{n!} [f(S \cup \{i\}) - f(S)], \tag{6}$$

where $f(S)$ denotes the network output when we keep variables in $S$ unchanged while masking variables in $N \setminus S$ by following the setting in Ancona et al. (2019). It has been proven that the Shapely value is a unique method to fairly allocate overall reward to each player that satisfies *linearity*, *nullity*, *symmetry*, and *efficiency* properties.

**Connections between the Shapley interaction index and the Shapely value.** Input variables of a DNN usually interact with each other, instead of working individually. Based on the Shapley value, Grabisch & Roubens (1999) further proposes the Shapley interaction index to measure the interaction utility between input variables. The Shapley interaction index is the only axiomatic extension of the Shapley value, which satisfies *linearity*, *nullity*, *symmetry*, and *recursive* properties. For two variables $i, j \in N$, the Shapley interaction index $I(i, j)$ can be considered as the change of the numerical importance of variable $i$ by the presence or absence of variable $j$.

$$I(i, j) = \tilde{\phi}(i)_{j \text{ always present}} - \tilde{\phi}(i)_{j \text{ always absent}}, \tag{7}$$

where $\tilde{\phi}(i)_{j\text{always present}}$ denotes the Shapley value of the variable $i$ computed under the specific condition that variable $j$ is always present. $\tilde{\phi}(i)_{j \text{ always absent}}$ is computed under the specific condition that $j$ is always absent.

**Connections between the multi-order interaction and the Shapley interaction index.** Based on the Shapley interaction index, Zhang et al. (2020) further defines the order of interaction, which represents the contextual complexity of interactions. It has been proven that the above Shapley interaction index $I(i, j)$ between variables $i, j$ can be decomposed into multi-order interactions as follows:

$$I(i, j) = \frac{1}{n - 1} \sum_{m=0}^{n-2} I^{(m)}(i, j). \tag{8}$$

## A.2 PROOF OF MULTI-ORDER INTERACTIONS

There we explain why $I^{(m)}(i, j)$ has no difference whether we include variables $\{i, j\}$ in $S$ or not. In the setting of (Zhang et al., 2020; Deng et al., 2021), $I^{(m)}(i, j)$ takes the following form:

$$I^{(m)}(i, j) = \mathbb{E}_{S \subseteq N \setminus \{i,j\}, |S|=m}[\Delta f(i, j, S)], \tag{9}$$

where $\Delta f(i, j, S) = f(S \cup \{i, j\}) - f(S \cup \{i\}) - f(S \cup \{j\}) + f(S)$ and $i, j \notin S$. While in our formulation, the order $m' = m + 2$ corresponds to the context $S' = S \cup \{i, j\}$. Now we denote our version of the multi-order interaction as $I'^{(m')}(i, j)$ with $\Delta' f(i, j, S)$ and aim to show that $I^{(m)}(i, j) = I'^{(m')}(i, j)$.

It is trivial to obtain that $f(S \cup \{i, j\}) - f(S \cup \{i\}) - f(S \cup \{j\}) + f(S) = f(S') - f(S' \setminus \{i\}) - f(S' \setminus \{j\}) + f(S' \setminus \{i, j\})$, which indicates that $\Delta f(i, j, S) = \Delta' f(i, j, S')$. Therefor, we can get $I^{(m+2)}(i, j) = I'^{(m')}(i, j)$.

## A.3 THEORETICAL DISTRIBUTIONS OF $F^{(m)}$

Fig. 4 depicts the theoretical distributions of $F^{(m)}$ for different $n$. Unlike Fig. 3 (a), the empty set $\emptyset$ is allowed as the input for DNNs. Apparently, when the number of variables $n$ is very large

$(n \geq 100)$, $F^{(m)}$ is only positive for $m/n \to 0$. For macromolecules such as proteins, the number of atoms is usually more than ten thousand. If the theorem in (Deng et al., 2021) that the strengths $\Delta W^{(m)}(i, j)$ of learning the $m$-order interaction is strictly proportional to $F^{(m)}$ holds, DNNs would be impossible to put attention to any middle-order interactions, which is proven to be critical for modeling protein-protein interactions (Liu et al., 2020; Das & Chakrabarti, 2021).

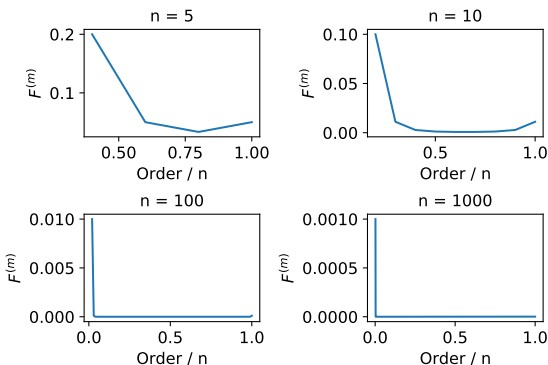

Figure 4: Distributions of $F^{(m)}$ with different numbers of variables $n$ where $f(\emptyset)$ is taken into consideration.

### A.4 EXPLANATION OF PROPOSITION 1

Prop. 1 illustrates an intuitive necessary condition for a model $f$ to achieve the global minimum loss. That is, the learned strength $J^{(m)}$ must match the optimal strength $J^{(m)*}$. Otherwise, $f$ must be inferior to the best predictor $f^*$. This proposition is very straightforward. The global minimum loss $L^*$ has its corresponding model weight $f^*$ and thereby a unique learned strength $J^{(m)*}$. Given another model $f$, if its strength $J^{(m)}$ is different from $J^{(m)*}$, then $f$ is different from $f^*$. As a consequence, the loss of $f$ must be larger than $L^*$. Notably, there we consider the loss function to be non-convex. In contrast, if the loss function is convex, standard optimization techniques like gradient descent will easily find parameters that converge towards global minima.

## B EXPERIMENTAL DETAILS AND ADDITIONAL RESULTS

### B.1 NEWTONIAN DYNAMICS DATASET

The following six forces are utilized in the dataset of Newtonian dynamics: (1) $1/r$ orbital force: $-m_1 m_2 \hat{r}/r$; (2) $1/r^2$ orbital force $-m_1 m_2 \hat{r}/r^2$; (3) charged particles force $q_1 q_2 \hat{r}/r^2$ (4) damped springs with $|r - 1|^2$ potential and damping proportional and opposite to speed; (5) discontinuous forces, $-\{0, r^2\}\hat{r}$, switching to 0 force for $r < 2$; and (6) springs between all particles, a $(r - 1)^2$ potential. There we only use the spring force for our experiments.

### B.2 TRAINING DETAILS

All experiments are implemented by Pytorch (Paszke et al., 2019) on an A100 GPU. An Adam (Kingma & Ba, 2014) optimizer is used without weight decay, and a ReduceLROnPlateau scheduler is enforced to adjust it with a factor of 0.6 and patience of 10. The initial learning rate is 1e-4, and the minimum learning rate is 5e-6. The batch size is 512 for the sake of a fast training speed. Each model is trained for 1200 epochs, and early stopping is used if the validation error fails to decrease for 30 successive epochs. We randomly split each dataset into training, validation, and test sets with a ratio of 80/10/10.

For both EGNN and Molformer, the numbers of layers (*i.e.*, depths) are 3, and the dimensions of the input feature are 32. Besides, Molformer has 4 attention heads and a dropout rate of 0.1. The dimension of the feed-forward network is 128. It is worth noting that we employ multi-scale

self-attention with a distance bar of $[0.8, 1.6, 3]$ to achieve better performance. This multi-scale mechanism helps Molformer to concentrate more on local contexts. However, it does not harm FC-graphs, and the connections between all pairs of entities remain. We also discover that the multi-scale mechanism has little impact on the distribution of $J^{(m)}$ and $J_D^{(m)}$. Regarding the setup of the ISGR algorithm, the threshold $\bar{J}$ to adjust the number of neighbors is tuned via a grid search. The interval of epochs $\Delta e$ is 10, and the initial $k_0 = 8$. Concerning baselines, we follow Klicpera et al. (2019) and Topping et al. (2021) and optimize hyperparameters by random search. Table 2 documents $\alpha$, $k$, and $\epsilon$ for DIGL, whose descriptions can be found in Klicpera et al. (2019). Table 3 reports the maximum iterations, $\tau$ and $C^+$ for SDRF, whose descriptions is available in Topping et al. (2021).

Table 2: Hyperparameters for DIGL.

| Task | Newtonian Dynamics | | Molecular Dynamics | | Hamiltonian Dynamics | | QM7 | | QM8 | |
|---|---|---|---|---|---|---|---|---|---|---|
| Model | EGNN | Molformer | EGNN | Molformer | EGNN | Molformer | EGNN | Molformer | EGNN | Molformer |
| $\alpha$ | 0.0259 | 0.1284 | 0.0732 | 0.1041 | 0.1561 | 0.3712 | 0.0655 | 0.2181 | 0.1033 | 0.1892 |
| $k$ | 32 | 32 | 32 | 32 | 64 | 64 | - | - | - | - |
| $\epsilon$ | - | - | - | - | 0.0001 | - | - | - | - | 0.0002 |

Table 3: Hyperparameters for SDRF.

| Task | Newtonian Dynamics | | Molecular Dynamics | | Hamiltonian Dynamics | | QM7 | | QM8 | |
|---|---|---|---|---|---|---|---|---|---|---|
| Model | EGNN | Molformer | EGNN | Molformer | EGNN | Molformer | EGNN | Molformer | EGNN | Molformer |
| Max Iter. | 15 | 11 | 39 | 34 | 16 | 13 | 22 | 17 | 25 | 12 |
| $tau$ | 120 | 163 | 54 | 72 | 114 | 186 | 33 | 35 | 48 | 60 |
| $C^+$ | 0.73 | 1.28 | 1.44 | 1.06 | 0.96 | 0.88 | 0.53 | 0.70 | 0.69 | 0.97 |

We create a simulated system with 10 identical particles with a unit weight for Hamiltonian and Newtonian cases. For QM7, QM8, and ISO17 datasets, we sample 10 molecules that have the lowest MAE. For Hamiltonian and Newtonian datasets, we sample 100 timeframes that have the lowest prediction errors. Then for each molecule or dynamic system, we compute all pairs of entities $i, j \in [N]$ without any sampling strategy. Moreover, we limit the number of atoms between 10 and 18 to compute the interaction strengths for QM7 and MQ8.

### B.3 EXAMINATION OF THE NORMAL DISTRIBUTION HYPOTHESIS

We use *scipy.stats.normaltest* in the Scipy package (Virtanen et al., 2020) to test the null hypothesis that $\frac{\partial \Delta f(i,j,S)}{\partial W}$ comes from a normal distribution, i.e, $\frac{\partial \Delta f(i,j,S)}{\partial W} \sim \mathcal{N}\left(0, \sigma^2\right)$. This test is based on D'Agostino and Pearson's examination that combines skew and kurtosis to produce an omnibus test of normality. The $p$-values of well-trained EGNN and Molformer on the Hamiltonian dynamics dataset are 1.97147e-11 and 2.38755e-10, respectively. The $p$-values of randomly initialized EGNN and Molformer on the Hamiltonian dynamics dataset are 2.41749e-12 and 9.78953e-07, separately. Therefore, we are highly confident in rejecting the null hypothesis (*e.g.*, $\alpha = 0.01$) and insist that $\frac{\partial \Delta f(i,j,S)}{\partial W}$ depends on the data distributions of downstream tasks and the backbone model architectures.

## C CNNS AND MLP-MIXER ON VISUAL TASKS

To investigate the change of interaction strengths during the training process in image classification, we train ResNet-50 (He et al., 2016) and MLP-Mixer (*Small*) (Tolstikhin et al., 2021) and calculate the interaction strength by the official implementation provided by Deng et al. (2021). MLP-mixer is an architecture based exclusively on multi-layer perceptrons (MLPs). It contains two types of layers: one with MLPs applied independently to image patches (*i.e.* "mixing" the per-location features), and one with MLPs applied across patches (*i.e.* "mixing" spatial information). We discuss MLP-mixer to compare it with the traditional CNNs. Notably, CNNs assume the local inductive bias, while MLP-mixer instead connects each patch with other patches (*e.g.*, no constraint of locality).

We follow the training settings of DeiT (Touvron et al., 2021) and train 200 epochs with the input resolution of $224 \times 224$ on CIFAR-10 (Krizhevsky et al., 2009) and ImageNet (Russakovsky et al., 2015) datasets. Fig. 3 plots the corresponding strengths at different epochs, where the dotted line denotes the initial interaction strength without training (referring to epoch 0), *i.e.*, the data distribution of strengths $J_D^{(m)}$.

Through visualization, it can be easily found that $J_D^{(m)}$ in both CIFAR-10 and ImageNet have already obeyed a mode that the low-order ($m/n \leq 0.2$) and high-order ($m/n \geq 0.8$) interaction strengths are much higher than middle-order ($0.2 \leq m/n \leq 0.8$). The variation of interaction strengths is very slight with the training proceeding, which validates our statement that the data distribution has a strong impact on the learned distribution. More importantly, we challenge the argument in Deng et al. (2021), who believe it is difficult for DNNs to encode middle-order interaction. But in our experiments on GNNs, we document that DL-based models are capable of capturing middle-order interactions.

The capability of CNNs to capture desirable levels of interactions is constrained by improper inductive bias. Remarkably, some preceding work (Han et al., 2021; Ding et al., 2022) has proved the effectiveness of enlarging kernel size to resolve the local inductive bias, where adequately large kernel size is able to improve the performance of CNNs comparable to ViT and MLP-Mixer. Nevertheless, how determining the scale of convolutional kernels is still under-explored. Our ISGR algorithm provides a promising way to seek the optimal kernel size based on the interaction strength, abandoning the exhaustive search.

### C.1 GRAPH CONSTRUCTION APPROACHES

Unlike social networks or knowledge graphs, there are, indeed, no explicit edges in graphs of most scientific problems. So in order to represent molecules or systems as graphs, *KNN-graphs* (see Fig. 5 (a)), *fully-connected graphs* (see Fig. 5 (b)), and *r-ball graphs* are the three most broadly used mechanisms to build the connectivity. In fact, FC-graphs is a special type of KNN-graphs, where $K \geq n - 1$. However, FC-graphs or KNN-graphs with a large $K$ suffer from high computational expenditure and are usually infeasible with thousands of entities. In addition, they are sometimes unnecessary since the impact from distant nodes is so minute to be ignored.

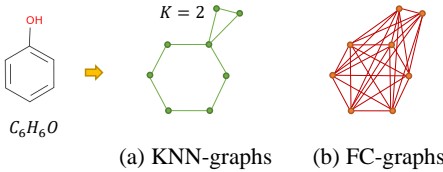

(a) KNN-graphs     (b) FC-graphs

Figure 5: Different graph constructions of the compound $C_6H_6O$.

## D COMPREHENSIVE COMPARISON TO EXISTING BOTTLENECKS OF GNNs

GNNs based on the message-passing diagram show extraordinary results with a small number of layers. Nevertheless, such GNNs fail to capture information that depends on the entire structure of the graph and prevent the information flow from reaching distant nodes. This phenomenon is called **under-reaching** (Barceló et al., 2020). To overcome this limitation, an intuitive resolution is to increase the layers. But unfortunately, GNNs with many layers tend to suffer from the **over-smoothing** (Oono & Suzuki, 2019) or **over-squashing** (Alon & Yahav, 2020) problems.

*Over-smoothing* takes place when node embeddings become indistinguishable. It occurs in GNNs that are used to tackle short-range tasks, *i.e.*, the accurate prediction majorly depends on the local neighborhood. On the contrary, long-range tasks require as many layers as the range of interactions between nodes. But this would contribute to the exponential increase of the node's receptive field and compress the information flow, which is named *over-squashing*. In our study, we do not specify which category of problems to be addressed (*i.e.*, long-range or short-range). Instead, we aim to explore which sort of interactions that GNNs are more likely to encode (*i.e.*, too simple, intermediately

complex, and too complex). It is also worth noting that different tasks require different levels of interactions. For instance, Newtonian and Hamiltonian dynamics demand too complex interactions, while molecular property prediction prefers interactions of intermediate complexity. Then based on both theoretical and empirical evidence, we discover that improper inductive bias introduced by the way to construct graph connectivity prevents GNNs from capturing the desirable interactions, resulting in the representation bottleneck of GNNs.

So what is the significant difference between our representation bottleneck and *over-squashing*? Most foundationally and importantly, our representation bottleneck is based on the theory of multi-order interactions, while *over-squashing* relies on the message propagation of node representations. To be specific, it is demonstrated in Alon & Yahav (2020) that the propagation of messages is controlled by a suitable power of $\hat{A}$ (the normalized augmented adjacency matrix), which relates *over-squashing* to the graph topology. In contrast, we show that the graph topology strongly determines the distribution of interaction strengths $J^{(m)}$, *i.e.*, whether GNNs are inclined to capture too simple or too complex interactions. This difference in theoretical basis leads to the following different behaviors of our representation bottleneck and *over-squashing*:

- The multi-order interaction technique focuses on interactions under a certain context, whose complexity is measured as the number of its variables (*i.e.*, nodes) $m$ divided by the total number of variables of the environment (*i.e.*, the graph) $n$. Thus, the complexity of interactions is, indeed, a relative quantity. Conversely, *over-squashing* (as well as *under-reaching*) concerns about the absolute distance between nodes. Given a pair of nodes $i$ and $j$, if the shortest path between them is $r$, then at least $r$ layers are required for $i$ to reach out to $j$. More generally, long-range or short-range tasks discussed in most GNN studies are referring to this $r$-distance. *Over-squashing*, therefore, follows this $r$-distance metric and argues that the information aggregated across a long path is compressed, which causes the degradation of GNNs' performance.

  As a result, our representation bottleneck can occur in both short-range and long-range tasks, but *over-squashing* mainly exists in long-range problems. For short-range tasks, if we assume a KNN-graph with a large $K$ or even fully-connected graphs (*i.e.*, nodes can have immediate interactions with distant nodes), then the receptive field of each node is very large and GNNs intend to concentrate on too complex interactions but fail to capture interactions within local neighbors. For long-range tasks, if we assume a KNN-graph with a small $K$ (*i.e.*, nodes only interact with nearby nodes), then the receptive field of each node is relatively small compared to the size of the entire graph. Consequently, GNNs prefer to capture too simple interactions but are incapable of seizing more informative complex interactions.

- More essentially, the multi-order interaction theory of our representation bottleneck is model-agnostic, but the starting point of *over-squashing* is message passing, the characteristic of most GNN architectures. To make it more clear, the calculation of multi-order interactions (see Equ. 1 and 3) is completely independent of the network (*e.g.*, CNNs, GNNs, RNNs). However, the theory of *over-squashing* is founded on the message-passing procedure. This hypothesis makes *over-squashing* limited to the group of GNNs that are built on message passing. But other kinds of GNNs such as the invariants of Transformers may not suffer from this catastrophe. Instead, the analysis of multi-order interactions in our representation bottleneck can be utilized in any GNN architecture, even if it abandons the traditional message-passing mechanism.

To summarize, our representation bottleneck is more universal than *over-squashing*, which is built upon the absolute distance and merely talks about long-range tasks. This is due to the fact that our representation bottleneck is given birth to by the theory of multi-order interactions rather than the property of message propagation.

## E MORE RELATED WORK

**Expressiveness of GNNs.** It is well-known that MLP can approximate any Borel measurable function (Hornik et al., 1989), but few study the universal approximation capability of GNNs (Wu et al., 2020). Hammer et al. (2005) demonstrates that cascade correlation can approximate functions with structured outputs. Scarselli et al. (2008a) prove that a RecGNN (Scarselli et al., 2008b) can approximate any function that preserves unfolding equivalence up to any degree of precision. Maron et al. (2018) show that an invariant GNN can approximate an arbitrary invariant function defined

on graphs. Xu et al. (2018) show that common GNNs including GCN (Kipf & Welling, 2016) and GraphSage (Hamilton et al., 2017) are incapable of differentiating different graph structures. They further prove if the aggregation functions and the readout functions of a GNN are injective, it is at most as powerful as the Weisfieler-Lehman (WL) test (Leman & Weisfeiler, 1968) in distinguishing different graphs.

**GNNs' representation capacity.** It becomes an emerging area to evaluate the representation capability of DNNs (Shwartz-Ziv & Tishby, 2017; Neyshabur et al., 2017; Novak et al., 2018; Weng et al., 2018; Fort et al., 2019). Interactions between variables are pioneeringly to inspect the limitation of DNNs in feature representations (Zhang et al., 2020; Deng et al., 2021). Notwithstanding, prior works merely highlight the behaviors of general DNNs and examine their assertions via MLP and CNNs. In comparison, we emphasize GNNs that operate on structured graphs, distinct from images and texts.

