# OpenReview forum: "Representation Bottleneck of Graph Neural Networks for Scientific Problems"
_ICLR.cc/2024/Conference — Submitted to ICLR 2024_

### Official Review · Reviewer_WXTK · 2023-10-30

**Soundness:** 2 fair
**Presentation:** 2 fair
**Contribution:** 2 fair
**Rating:** 3
**Confidence:** 4

**Summary:**

The paper aims to explore the capacity of GNNs to model complex interactions. Using some specific measures, the authors argue that GNNs are unable to capture some interactions. To amend this, the authors propose a new rewiring scheme that aims to tackle this problem.

**Strengths:**

The question that the paper studies is interesting and relevant as it is related to existing literature on over-squashing. Focusing on geometric domains such as ones induced by some underlying physical model is an interesting motivation as such data is highly structured and it may be easier to find meaningful trends when considering interactions within the domain. The way in which the interactions are measured is interesting -- although, from my understanding, not novel.

**Weaknesses:**

While the paper has promise, it has some limitations.

(W1) Overall I found the structure of the paper and the wording to be a bit confusing. It would be helpful to add some summary sections or a summary of the structure of the paper at the start of the work to help the reader follow. To me, it is also not immediately clear what the contributions of the paper are. There are also typos, broken reference links, and strange/uncommon usage of words in the work although this critique might be considered more cosmetic.

(W2) Proposition 1 to me seems trivial and true by definition of a global/local minimum.

(W3) The figures are very small and hard to read.

**Questions:**

Would it be possible to provide a clear list of the contributions of the work? My understanding is that the measure of interactions is not novel, which instead could be an interesting contribution of the paper.

Would it be possible to provide a clear summary of each section of the paper, at the moment I am finding it difficult to follow the point of each section.

How do you compute for the data distribution $J_D^{(m)}$? My understanding is tha $J$ is computed through a model $f$, which would not be available for the "data" distribution?

In your rewiring algorithm -- do you only consider building k-NN graphs? The point is then that you are proposing a way to choose $k$ based on the measure $J$. Am I understanding correctly?

---

### Official Review · Reviewer_hyyD · 2023-10-30

**Soundness:** 2 fair
**Presentation:** 3 good
**Contribution:** 3 good
**Rating:** 5
**Confidence:** 4

**Summary:**

This paper investigates GNNs through the lens of interaction strength. The paper discovers that different inductive biases will cause different sensitivity of interaction strength orders. Under the understanding of interaction strength, the authors are motivated by CNN's observation to adjust KNN's K for interaction strength matching. The experiments demonstrate the discovered inductive biases/interaction strength alignment, the performance of the proposed adjustment, and the inductive bias comparisons between GNNs and DNNs.

**Strengths:**

1. Overall, this is an interesting paper that introduces inductive bias in GNNs using interaction strengths.
2. The authors discovered that the interaction strength mismatching is one of the important reasons GNNs can be trapped in a local minimum.
3. The paper also introduces how different inductive biases: KNN-graphs (with EGNN) and FC-graphs (with Malformed) affect the interaction strength.
4. According to experimental observations, the authors propose an effective algorithm to adjust inductive bias using KNN-graphs to achieve better results.
5. Authors further discuss inductive bias in DNNs.
6. The presentation of the paper is generally good.

**Weaknesses:**

**The paper includes so many aspects that the authors fail to cover each of them in-depth.**

1. I hope to see more justifications in terms of the new metric introduced for node-level multi-order interaction.
2. The paper has the potential to be a great empirical study. However, the experiments to show the representation bottleneck are too limited.
3. The proposed method is not well-motivated. Although the authors claim that the method is motivated by a tendency during the experiments, I still think the method is not well-motivated. For example, why do the authors adopt rewiring? Why did the authors choose KNN? And how and why is the threshold $\bar{J}$ determined?
4. The soundness of the method is also a concern. I did not see any theoretical insights provided to justify why the method works.
5. The performance improvement in regards to Molformer is limited.

**Questions:**

1. I'm confused why for two inductive biases KNN-graphs and FC-graphs, only one method for each was compared. I would like to see more methods for each of KNN-graphs and FC-graphs in Figure 1 to see the significance of the phenomenon.
2. In terms of graph structure learning, how do those methods affect the interaction strength?
3. The authors mentioned the proposed method can achieve stronger interpretability and generalization, but I failed to find the corresponding evidence. In terms of generalization ability, the authors may need to compare those methods with generalization design and dive into how their designs affect the interaction strength.

Overall, I feel the paper has an interesting core but the authors choose to cover it superficially from many aspects instead of digging into one of them. This makes the claims in this paper not strong and clear enough.

---

### Official Review · Reviewer_wmgx · 2023-11-01

**Soundness:** 3 good
**Presentation:** 3 good
**Contribution:** 3 good
**Rating:** 6
**Confidence:** 3

**Summary:**

This paper investigates the representation bottleneck of graph neural networks (GNNs) for scientific problems. GNNs rely on the message-passing paradigm to propagate node features and build interactions, but different graph learning problems require different ranges of node interactions. The authors propose a novel graph rewiring approach to address this issue, which adjusts the receptive fields of each node dynamically. The effectiveness of the algorithm is demonstrated on real-world and synthetic datasets. The paper also provides supplementary materials on the multi-order interaction tool and proves the equivalence of the reformed multi-order interaction to the original definition. Overall, the paper's contributions include identifying the limitations of GNNs in capturing node interactions, proposing a novel graph rewiring approach to address this issue, and providing supplementary materials and proofs for the multi-order interaction tool.

**Strengths:**

This paper makes several significant contributions to the field of graph neural networks (GNNs) for scientific problems.

Originality:
The paper proposes a novel graph rewiring approach to address the representation bottleneck of GNNs, which adjusts the receptive fields of each node dynamically. This approach is a creative combination of existing ideas and provides a new solution to the problem of capturing node interactions in GNNs. The paper also introduces a metric of multi-order interactions for GNNs from both node-level and graph-level perspectives, which is a new definition in the field.

Quality:
The paper is well-written and provides clear explanations of the proposed approach and the multi-order interaction tool. The experiments are conducted on real-world and synthetic datasets, and the results demonstrate the effectiveness of the proposed approach. The paper also provides supplementary materials and proofs for the multi-order interaction tool, which enhances the quality of the paper.

Clarity:
The paper is well-organized and easy to follow. The authors provide clear explanations of the proposed approach and the multi-order interaction tool, and the supplementary materials are helpful for readers who are not familiar with this field. The experiments are also well-designed and easy to understand.

Significance:
The paper's contributions are significant for the field of GNNs for scientific problems. The proposed graph rewiring approach provides a new solution to the problem of capturing node interactions in GNNs, which has important implications for various scientific domains. The multi-order interaction tool is also a valuable contribution to the field, as it provides a new metric for evaluating the performance of GNNs. Overall, the paper's contributions have the potential to advance the field of GNNs and have important implications for various scientific domains.

**Weaknesses:**

While the paper makes several significant contributions to the field of graph neural networks (GNNs) for scientific problems, there are also some weaknesses that could be addressed to improve the paper:

1. Lack of comparison with state-of-the-art methods: The paper does not compare the proposed approach with state-of-the-art methods for capturing node interactions in GNNs. It would be helpful to include such a comparison to demonstrate the effectiveness of the proposed approach.

2. Limited scope of experiments: The experiments are conducted on a limited number of real-world and synthetic datasets. It would be helpful to include more datasets to demonstrate the generalizability of the proposed approach.

3. Lack of analysis of computational complexity: The paper does not provide an analysis of the computational complexity of the proposed approach. It would be helpful to include such an analysis to demonstrate the scalability of the proposed approach.

4. Lack of discussion on limitations: The paper does not discuss the limitations of the proposed approach. It would be helpful to include a discussion on the limitations of the proposed approach and potential directions for future research.

5. Lack of clarity in some parts: While the paper is generally well-written and easy to follow, there are some parts that could be clarified further. For example, the introduction of the multi-order interaction tool could be explained more clearly for readers who are not familiar with this field.

Overall, addressing these weaknesses could improve the paper and make its contributions more impactful.

**Questions:**

1. Can you provide a comparison of the proposed approach with state-of-the-art methods for capturing node interactions in GNNs? This would help demonstrate the effectiveness of the proposed approach.

2. Can you include more datasets in the experiments to demonstrate the generalizability of the proposed approach? The experiments are currently conducted on a limited number of real-world and synthetic datasets.

3. Can you provide an analysis of the computational complexity of the proposed approach? This would help demonstrate the scalability of the proposed approach.

4. Can you discuss the limitations of the proposed approach and potential directions for future research? This would help provide a more complete picture of the proposed approach and its implications for future research.

5. Can you clarify some parts of the paper, such as the introduction of the multi-order interaction tool, for readers who are not familiar with this field? This would help make the paper more accessible to a wider audience.

6. Have you considered the potential impact of the proposed approach on the interpretability of GNNs? The dynamic adjustment of receptive fields could make it more difficult to interpret the learned representations, and it would be helpful to discuss this potential limitation.

7. Have you considered the potential impact of the proposed approach on the training stability of GNNs? The dynamic adjustment of receptive fields could make the training process more unstable, and it would be helpful to discuss this potential limitation and any strategies to address it.

---

### Meta-Review · Area_Chair_RFax · 2023-12-08

**Metareview:**

The paper focuses on addressing the representation bottleneck of graph neural networks (GNNs) through a novel graph rewiring approach that dynamically changes the interaction scope of each node. The introduction of a multi-order interaction tool from both node-level and graph-level perspectives is also a novel and useful metric in GNN research. However, the paper falls short in several areas, including the depth of analysis, theoretical justification, and clarity in presentation. The reviewers suggest improvements in comparing with other state-of-the-art methods, expanding the scope of experiments, providing more theoretical insights, and improving the overall presentation for a more comprehensive and compelling submission. There lacks a author response to address these concerns and questions during the rebuttal. Thus, a rejection is recommended.

**Justification For Why Not Higher Score:**

No author rebuttal provided.

**Justification For Why Not Lower Score:**

N/A

---

### Decision · Program_Chairs · 2024-01-16

Reject